# Fine-Mapping Array Design for Multi-Ethnic Studies of Multiple Sclerosis

**DOI:** 10.3390/genes10110903

**Published:** 2019-11-07

**Authors:** Ashley H. Beecham, Jacob L. McCauley

**Affiliations:** 1John P. Hussman Institute for Human Genomics, Miller School of Medicine, University of Miami, Miami, FL 33136, USA; abeecham@med.miami.edu; 2John T. Macdonald Foundation Department of Human Genetics, Miller School of Medicine, University of Miami, Miami, FL 33136, USA

**Keywords:** multiple sclerosis, fine-mapping, association

## Abstract

While approximately 200 autosomal genetic associations outside of the major histocompatibility complex (MHC) have been identified for multiple sclerosis (MS) risk in European populations, causal variants identified at the majority of these associated loci have been much more elusive. We propose that knowledge gained from replication efforts in Hispanic and African American populations can be utilized to more efficiently fine-map these risk loci. To this end, we have customized a genotyping array by adding ~20,000 bead types (~17,000 variants) to the base content of the Ilumina Infinium expanded multi-ethnic genotyping array and the Infinium ImmunoArray-24 v2 BeadChip. These custom bead types were chosen to allow for the detection of causal variation (1) in the presence of allelic and locus heterogeneity, by incorporating regulatory and coding variation within 1-Mb of previously identified risk variants and (2) in the absence of allelic and locus heterogeneity by incorporation of variants using linkage disequilibrium criteria, which are based on knowledge of replication status in Hispanic and African American study samples. This array has been designed to maximize fine-mapping potential for currently identified MS susceptibility loci, particularly in multi-ethnic populations. The strategies described here could be additionally informative for fine-mapping of other disease phenotypes.

## 1. Introduction

Although there has been great advancement in the discovery of genetic variations that are associated with multiple sclerosis (MS) risk in European populations (currently accounting for ~39% of the narrow-sense heritability) [1], we have yet to identify the underlying causal variation at the majority of these associated loci. This is not altogether unanticipated since the discovery effort has primarily made use of genome-wide arrays, which are designed to tag causal variants by targeting common variants across the genome [2,3,4]. In order to pinpoint variants that are truly consequential to MS pathology, the regions surrounding the associated variants must be narrowed. Several strategies exist for this process, known as fine-mapping, including heuristics based on linkage disequilibrium patterns (LD), penalized regression, and a Bayesian methodology [5].

One such fine-mapping effort was undertaken among European populations in 2013 by the International Multiple Sclerosis Genetics Consortium (IMSGC) [6]. A total of 14,498 individuals with MS and 24,091 controls were genotyped on the HumanImmuno v1 BeadChip (also known as the “ImmunoChip”), designed to deeply interrogate susceptibility loci that had previously been identified for 12 autoimmune diseases [7]. At the time of the array design in 2011, the MS research community contributed a total of 26 susceptibility loci that had been identified for MS risk, utilizing genome-wide association studies (GWAS). Due to the overlap in genetic architecture across autoimmune diseases, a total of 40 of the loci included in the design had been determined to be associated with MS through additional GWAS by the time that genotyping on the ImmunoChip array was completed. Subsequent analysis of the array resulted in 97 independent non-major histocompatibility complex (MHC) associations with MS risk (93 primary signals and four secondary, resulting from stepwise conditional analysis at each fine-mapping locus).

Based on imputation quality within a 2-Megabase (Mb) isolated window surrounding each variant (1-Mb on each side), only 68 of the 97 association signals were determined to be eligible for fine-mapping. Using a Bayes Factor to measure the posterior probability that any single variant drives association, only eight of 68 were then fine-mapped to high resolution; meaning that there were at most five variants containing 50% of the posterior probability and that the top associated variant was in the smallest credible SNP set containing 90% of the posterior probability [6]. The number of independent MS association signals identified outside of the MHC has now grown from 97 to 200 due to the efforts of the IMSGC [1], yet no further large-scale fine-mapping efforts have been undertaken. While some progress has been made towards the fine-mapping of MS risk associations, there is still much headway needed in order to fine-map all 200 independent variants, which span 156 loci (2-Mb each in size, i.e., extending 1-Mb on each side of the primary risk variant) [1].

In our previous replication study in Hispanics and African Americans [8], we found that not all of the 200 currently identified variants were replicated among these minority populations. In fact, less replication was found than expected, particularly among African Americans. We noted several possible explanations including: (1) that the smaller LD blocks observed in African Americans could result in less correlation between the analyzed and causal variants, inherently reducing power for detection, (2) locus heterogeneity, or (3) allelic heterogeneity. We propose that we can leverage the information that was gained from this replication effort and the relatively smaller LD blocks observed among both Hispanics and African Americans in order to more efficiently fine-map these risk loci. An ImmunoChip study in an African American sample of 803 individuals with MS and 1516 controls additionally supported the utility of African Americans in localizing risk signals, finding several loci that showed an alternative but correlated SNP in the same region as Europeans [9].

Due to the high costs of next-generation sequencing and the power needed for variant detection, we therefore sought to design a customized array in order to cost-efficiently fine-map these 200 MS risk associations in multi-ethnic populations. The base content of this custom array includes the Ilumina Infinium expanded multi-ethnic genotyping array (MEGA^EX^) and the Infinium ImmunoArray-24 v2 BeadChip (ImmunoChipV2); to which we have added additional custom content of ~20,000 bead types or ~17,000 variants (Figure 1). The base array content allows for both substantial coverage of many previously identified autoimmune loci, although the ImmunoChipV2 was designed primarily for European populations, in addition to a robust genome-wide platform with great imputation capability. The genome-wide coverage of the MEGA^EX^ rivals that of the larger Illumina Omni 2.5 array, but specifically leverages the identification of novel variation (identified through whole genome sequencing) as a means to more fully capture and tag more moderate-frequency genetic variation within genetically diverse minority populations (i.e., African Americans and Hispanics). The genome-wide coverage may additionally be useful for quality control, ancestry estimation, and additional downstream analyses. The custom content was designed using a two-pronged approach with both the potential presence and absence of allelic and locus heterogeneity in mind.

## 2. Coding and Regulatory Variation

We designed the custom content to allow for detection of causal variation in the presence of allelic and locus heterogeneity by incorporating regulatory and coding variation spanning 1-Mb from the primary SNP in each of the 156 regions and within the MHC (27–34 Mb on chromosome 6). In order to prioritize variation, we used several filters; the first of which was a frequency filter in order to facilitate inclusion of variants for which we would have power to detect association. We identified a total of 2,232,034 unique variants from the Genome Aggregation Database (gnoMAD) [10], Exome Aggregation Consortium (ExAC) [11], and 1000 genomes [12] with frequencies as follows: Africans (AFR) minor allele frequency (MAF) > 0.01, Hispanics (AMR) MAF > 0.01, non-Finnish Europeans (NFE) MAF > 0.05, or Eastern Asians (EAS) MAF > 0.05. A lower-frequency threshold was included for the AFR and AMR groups since these have been our recent populations of research focus. After excluding variations that overlapped with the base array content, 2,017,303 variants remained.

The second filter prioritized coding variants by utilizing ANNOtate VARiation(ANNOVAR) software [13] for inclusion of exonic (5424 non-synonymous or stop-gain/loss variants) and splice (83 variants) sites, and prioritized regulatory variants by utilizing combined annotation-dependent depletion (CADD) [14] and RegulomeDB [15]. We included variants with a CADD score of ≥30 (418 variants), indicating that each variant is in the top 0.1% of deleterious variants in the human genome. Unlike most alternative scoring algorithms, which rely on a single annotation type or are restricted in scope, CADD integrates multiple annotations spanning both conservation and function. We additionally included variants with a RegulomeDB score of 1a–1e (883 variants), indicating that the variant is an expression quantitative trait locus (eQTL) and at a transcription factor binding site. We acknowledge that there are additional databases that could have been used to identify eQTLs, including the Genotype-Tissue Expression (GTEx) portal [16], and so our variant list is likely to be conservative. After excluding tri-allelic variants that pose potential design challenges, we were left with a total of 6518 coding or regulatory variants.

## 3. Fine-Mapping Content

We additionally designed the custom content to allow for detection of causal variation in the absence of allelic and locus heterogeneity by leveraging information regarding the replication status of each of the 200 autosomal non-MHC variants in our Hispanic and African American samples [8]. We extracted 3,207,498 variants that were within 1-Mb of each of the 200 variants from 1000 Genomes Project populations (phase 3, version 5) of unrelated non-Finnish Europeans (NFE: 403 individuals of northern and western European ancestry in Utah (CEU), Toscani in Italy (TSI), British in England and Scotland (GBR), and Iberians in Spain (IBS)); African Americans (AA: 149 individuals of African ancestry in southwestern United States (ASW), and African Caribbeans in Barbados (ACB)); and Hispanics (AMR: 338 individuals of Mexican ancestry in Los Angeles (MXL), Puerto Ricans in Puerto Rico (PUR), Colombians in Colombia (CLM), and Peruvians in Peru (PEL)).

These variants were then filtered further using LD criteria, which were specific to replication statuses based on the previous replication study [8]. For the 16 variants that were replicated in both the Hispanic and African American samples (one-sided *p* ≤ 0.05), the square of the correlation coefficient (R^2^) ≥ 0.1 with the associated variant was required from 1000 Genomes’ NFE, AMR, and AA samples. The presence of association at a shared variant across all populations indicated that the causal variant must also be in LD with this associated variant in all populations. A total of 916 variants met these LD criteria for the 16 variants. For the 85 variants that were only replicated in one study sample, we likewise required *R*^2^ ≥ 0.1 in 1000 Genomes samples from NFE and the replicating population, resulting in 12,640 variants. If the lack of association in the non-replicating population is due to limited LD between the presently analyzed and true causal variant, fine-mapping of the 12,640-variant subset in this population may in fact be ideal as it may result in a smaller credible interval. For the five variants that showed significant yet opposite directions of effect (two-sided *p* ≤ 0.10) in one study sample, we required *R*^2^ ≥ 0.1 in 1000 Genomes samples from NFE or the relevant population, given this evidence for variation in LD structures. This resulted in 2105 variants. For the remaining 94 variants that did not replicate in either study sample, no further filtering could be done beyond *R*^2^ ≥ 0.1 in 1000 Genomes’ NFE samples, with 23,246 variants meeting this criterion. After removal of overlapping base content, 28,295 unique variants remained. Additionally, removing tri-allelic variants resulted in a total of 28,238 variants.

Additional fine-mapping content for the MHC was not included due to space limitations, although the ImmunoChipV2 provides fairly ample coverage. In addition, the MEGA^EX^ contains a panel of variants dedicated to imputing classical human leukocyte antigen (HLA) alleles, providing for a more comprehensive tool to aid our fine-mapping and novel-variant discovery of this region within our admixed Hispanic and African American cohorts. Between the MEGA^EX^ and ImmunoChipV2 array, ≥90% of the variants used to impute classical HLA alleles (A, B, C, DPB1, DQA1, DQB1, and DRB1) with HLA Genotype Imputation with Attribute Bagging (HIBAG) [17] are included. Fine-mapping of this region in Hispanics and African Americans has also been the focus of several currently unpublished efforts by ourselves as well as others (personal communication) utilizing the ‘MS Chip’ array [1], which was designed with this intent in mind. Nonetheless, the regulatory and coding variation included from this region provides the opportunity for further insight, particularly among minority populations.

## 4. Final Pruning

In total, 34,637 unique variants met the final filtering criteria using either approach. After using the Illumina iSelect design tool [18] for variant scoring based on design ability, 31,966 remained. To reduce this variant set further, we prioritized the 15,107 variants that were obtained using the regulatory/coding approach or met the LD criteria for the 101 variants that were replicated in either study sample. The remaining 16,859 variants were pruned using an imputation-based approach. A subset of 95 individuals from the Hispanic study sample were genotyped on the MEGA^EX^ array as part of a pilot project. We imputed these 95 individuals using 1000 Genomes (phase 3, version 5) data, and then excluded the variants that had subsequent imputation information quality (INFO) scores > 0.857, as we would anticipate being able to impute these fairly well using the base content of our array. This INFO threshold was chosen arbitrarily to obtain the pre-selected number of bead types. After this pruning step, 1635 of the 16,859 variants remained. In total, 15,107 + 1635 = 16,742 variants (19,810 bead types) comprised the custom content.

## 5. Summary

This array has been designed to maximize fine-mapping potential for currently identified MS susceptibility loci, particularly in multi-ethnic populations. There is a historical lack of minority participant inclusion in genetic studies across various disease phenotypes, with only 4% of all published genome-wide studies reporting on non-European populations in 2009, increasing to 19% by 2016 [19,20,21]. Although researchers recognize the value of minority inclusion for the purpose of insight into health disparities, understanding biology, and improving clinical care, this imbalance persists [22].

This inconsistency could be due to limited participant engagement, making it difficult to attain the sample sizes necessary to achieve adequate power. This is true in Hispanic and African American MS study samples (with sizes currently approaching ~2000 cases and ~2000 controls in the Alliance for Research in Hispanic MS (ARHMS)) [8]; while European MS study samples currently approach 110,00 in the IMSGC (~50,000 MS cases and ~60,000 controls) [1]. Yet, the admixture and unique LD structure of Hispanics and African Americans make them distinctly valuable for localizing association signals and identifying causal variants, despite the limited sample size. This identification of causal variants is ultimately necessary for our understanding of MS pathology and subsequent improvement in drug targets. The strategies described here will enable cost-efficient fine-mapping in multi-ethnic MS populations, and could additionally be informative for fine-mapping array design efforts in other disease phenotypes.

## Figures and Tables

**Figure 1 genes-10-00903-f001:**
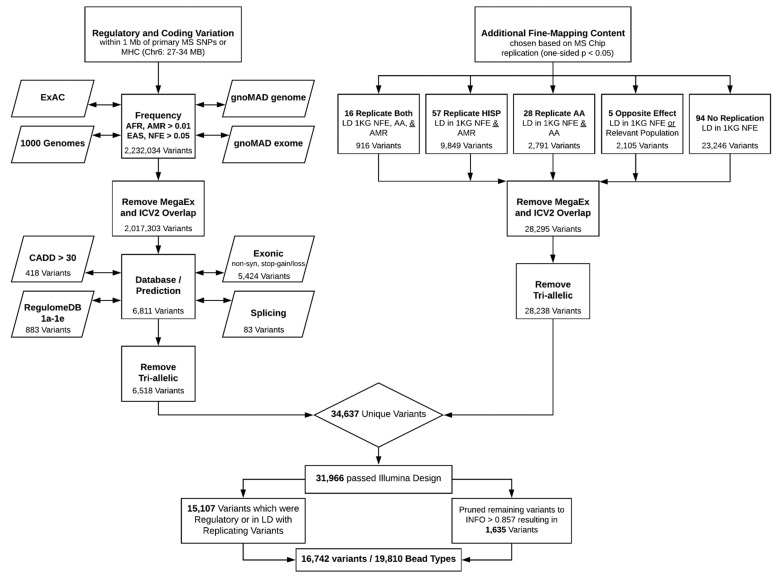
Workflow of variant selection for custom content.

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
