# Peer review of "Fine-Mapping Array Design for Multi-Ethnic Studies of Multiple Sclerosis"

_genes, 2019, doi:10.3390/genes10110903_

Round 1

Reviewer 1 Report

This communication reports the design of a fine-mapping array for its use in the analysis of multi-ethnic genetic associations for MS risk.  This methodology was carefully developed with the idea of further advancing previous studies from the authors indicating that not all of the 200 currently identified independent MS risk association signals replicated for the Hispanic and African American populations. 

The importance of the proposed strategy lies in the possibility of achieving cost-efficient detailed mapping of MS susceptibility loci in multi-ethnic MS populations and its extended application to different disease phenotypes.

Author Response

This communication reports the design of a fine-mapping array for its use in the analysis of multi-ethnic genetic associations for MS risk.  This methodology was carefully developed with the idea of further advancing previous studies from the authors indicating that not all of the 200 currently identified independent MS risk association signals replicated for the Hispanic and African American populations. 

The importance of the proposed strategy lies in the possibility of achieving cost-efficient detailed mapping of MS susceptibility loci in multi-ethnic MS populations and its extended application to different disease phenotypes.

We thank the reviewer for their acknowledgement of the importance of this array design, both in fine-mapping of MS susceptibility loci in multi-ethnic populations and in providing a strategy which is applicable to other disease phenotypes.

Reviewer 2 Report

The authors present results from designing a fine-mapping array for previously identified SNP associations with multiple sclerosis (MS).  Specifically, the authors aimed to select target SNPs that maximized fine-mapping utility in diverse populations, with an emphasis on Hispanic and African American individuals.  There are multiple motivations for such an endeavor, including the relatively high costs of resequencing and the advantage of different LD patterns in subjects of diverse genetic ancestry (particularly African ancestry).  Follow-up trans-ethnic fine-mapping can help resolve the identification of causal variation as well as ethnicity-specific variants under conditions of locus and allelic heterogeneity.

To this end, the authors outline their strategy for selecting fine-mapping SNPs to add to the existing Illumina Multi-Ethnic genotyping array.  An emphasis was placed on protein-coding and regulatory variation using available functional annotation

Overall the paper is very well-written and provides a nice roadmap for designing a fine-mapping array with intent to genotype diverse populations.  I only have one major comment with respect to variant prioritization based on regulatory potential.

Major Comments:

I believe RegulomeDB uses fairly old information, particularly when it comes to eQTL status.  For example, I don’t think any recent results from GEUVADIS or GTEx eQTL analyses are included.  Of note, Afrasiabi et al. (Genome Medicine, 2019) highlight 3 eQTL SNPs in their analyses of MS risk variants based on GTEx data, of which 2 (rs12588969, rs12148050) have low RegulomeDB scores and no listed eQTL evidence in RegulomeDB.

Minor Comments:

Figure 1 was a bit blurry and hard to read, and would benefit from being higher resolution.

MEGAEX sometimes does not have the “EX” superscripted.

Author Response

The authors present results from designing a fine-mapping array for previously identified SNP associations with multiple sclerosis (MS).  Specifically, the authors aimed to select target SNPs that maximized fine-mapping utility in diverse populations, with an emphasis on Hispanic and African American individuals.  There are multiple motivations for such an endeavor, including the relatively high costs of resequencing and the advantage of different LD patterns in subjects of diverse genetic ancestry (particularly African ancestry).  Follow-up trans-ethnic fine-mapping can help resolve the identification of causal variation as well as ethnicity-specific variants under conditions of locus and allelic heterogeneity.

To this end, the authors outline their strategy for selecting fine-mapping SNPs to add to the existing Illumina Multi-Ethnic genotyping array.  An emphasis was placed on protein-coding and regulatory variation using available functional annotation. Overall the paper is very well-written and provides a nice roadmap for designing a fine-mapping array with intent to genotype diverse populations. 

We agree with the reviewer that there are many advantages to minority inclusion in fine-mapping efforts, and we are thankful for the appreciation of our strategy in this regard.

I only have one major comment with respect to variant prioritization based on regulatory potential.

Major Comments:

I believe RegulomeDB uses fairly old information, particularly when it comes to eQTL status.  For example, I don’t think any recent results from GEUVADIS or GTEx eQTL analyses are included.  Of note, Afrasiabi et al. (Genome Medicine, 2019) highlight 3 eQTL SNPs in their analyses of MS risk variants based on GTEx data, of which 2 (rs12588969, rs12148050) have low RegulomeDB scores and no listed eQTL evidence in RegulomeDB.

We acknowledge the limitation of using only one source for eQTL data (i.e. RegulomeDB). Array design and production has been completed at this stage, and so no further updates to the design can be made. We have now noted this as limitation in the manuscript on page 3, line 109-111 as We acknowledge that there are additional databases which could have been used to identify eQTLs, including GTEx [16], and so our variant list is likely conservative.”

Minor Comments:

Figure 1 was a bit blurry and hard to read, and would benefit from being higher resolution.

Figure 1 has now been updated to higher resolution.

MEGAEX sometimes does not have the “EX” superscripted.

All instances of MEGAEX have now been updated to include the superscript.

Reviewer 3 Report

Many high-impact editorials have highlighted the need for GWAS in non-European studies. GWAS in non-European studies are hence clearly warranted, and specific genotyping arrays such as this one would be helpful. Authors have strong reputation for genotyping and GWAS in the MS field, and have a track record of GWAS in non-European study populations. Although the study only describes the design and no results yet, availability of this design would be of interest to others working in the field of MS and other complex traits.

Additional questions:

Selection frequency filter: do variants have to meet filter in all study populations or in only one to be included? If they have to meet filter in all study populations, would an important subset of variants be missed?

Is any information available about how many of the 16,742 variants are (expected to be) working on this array?

Author Response

Many high-impact editorials have highlighted the need for GWAS in non-European studies. GWAS in non-European studies are hence clearly warranted, and specific genotyping arrays such as this one would be helpful. Authors have strong reputation for genotyping and GWAS in the MS field, and have a track record of GWAS in non-European study populations. Although the study only describes the design and no results yet, availability of this design would be of interest to others working in the field of MS and other complex traits.

We thank the reviewer for these comments and agree that this array may be of interest to others in the field. We are therefore eager to make our manuscript available to the research community.

Additional questions:

Selection frequency filter: do variants have to meet filter in all study populations or in only one to be included? If they have to meet filter in all study populations, would an important subset of variants be missed?

In order to be included, it was only necessary that a variant meet the frequency filter in one population (i.e. MAF > 0.01 in AFR or MAF > 0.01 in AMR or MAF > 0.01 in NFE or MAF > 0.05 in EUR). We would therefore not anticipate missing any variants which met one of the specified frequency thresholds. This has now been clarified in the manuscript on page 3, line 97 by changing the ‘and’ to ‘or’ in the statement which now reads Africans (AFR) minor allele frequency (MAF) > 0.01, Hispanics (AMR) MAF > 0.01, non-Finnish Europeans (NFE) MAF > 0.05, or Eastern Asians (EAS) MAF > 0.05.”

Is any information available about how many of the 16,742 variants are (expected to be) working on this array?

Genotyping of the array in Hispanics and African Americans has not yet been completed. While all 16,742 variants made it successfully onto the array, at this time we are unable to provide further information regarding their performance in genotyping.